# CAR🚗: Controllable AutoRegressive Modeling for Visual Generation

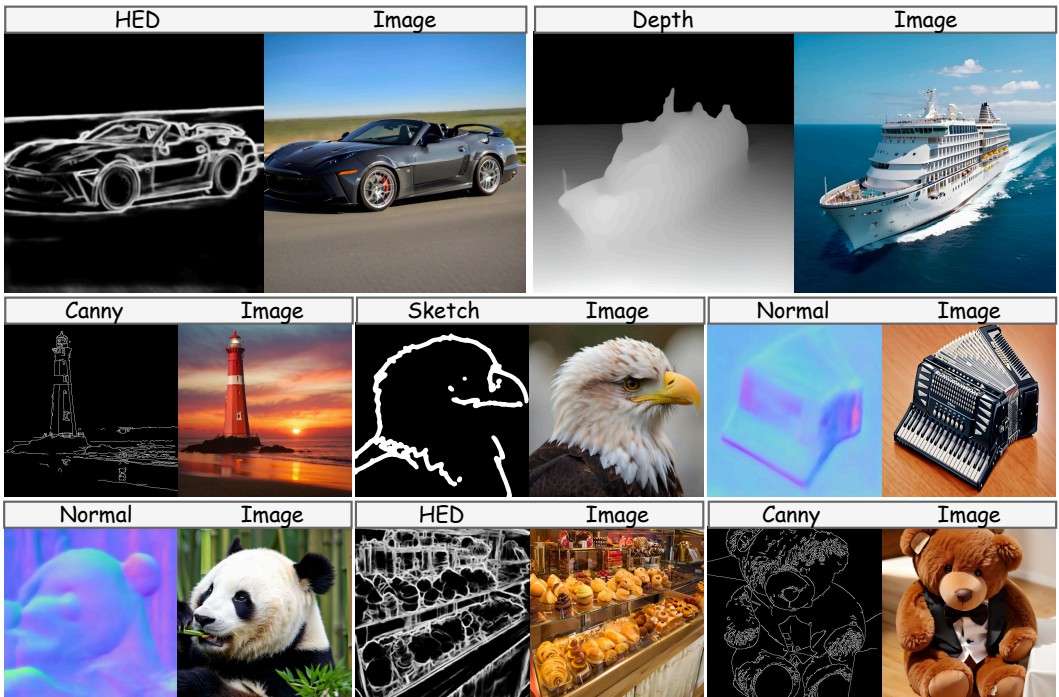

Figure 1: Controllable generation using **CAR** under various conditions, with each pair showing the condition on the left and the corresponding generated image on the right. Results are $512 \times 512$.

## Abstract

Controllable generation, which enables fine-grained control over generated outputs, has emerged as a critical focus in visual generative models. Currently, there are two primary technical approaches in visual generation: diffusion models and autoregressive models. Diffusion models, as exemplified by ControlNet and T2I-Adapter, offer advanced control mechanisms, whereas autoregressive models, despite showcasing impressive generative quality and scalability, remain underexplored in terms of controllability and flexibility. In this study, we introduce **C**ontrollable **A**uto**R**egressive Modeling (**CAR**), a novel, plug-and-play framework that integrates conditional control into multi-scale latent variable modeling, enabling efficient control generation within a pre-trained visual autoregressive model. CAR progressively refines and captures control representations, which are injected into each autoregressive step of the pre-trained model to guide the generation process. Our approach demonstrates excellent controllability across various types of conditions and delivers higher image quality compared to previous methods. Additionally, CAR achieves robust generalization with significantly fewer training resources compared to those required for pre-training the model. To the best of our knowledge, we are the first to propose a control framework for pre-trained autoregressive visual generation models.

## 1 INTRODUCTION

Controllable generation represents a pivotal aspect of visual generative models, enabling precise and fine-grained control over generated outputs. This capability is indispensable for tasks that demand a high degree of precision and adaptability, positioning it as a significant area of focus within the domain. Currently, there are mainly two primary paradigms that have substantially advanced the field of visual generation: diffusion models (Rombach et al., 2022; Saharia et al., 2022) and autoregressive models (Tian et al., 2024; Esser et al., 2021). While the diffusion paradigm has already given rise to numerous widely adopted methods for controllable generation, the autoregressive approach remains underexplored, particularly in how to empower the strengths of this paradigm for controllable generation, which constitutes the central emphasis of this work.

Diffusion models utilize iterative denoising processes based on Markov chains to produce high-quality outputs. These models have inspired the development of widely used controllable generation techniques such as ControlNet (Zhang et al., 2023) and T2I-Adapter (Mou et al., 2024), which can provide granular control over the generation images by incorporating additional signals such as edge maps and human poses. However, challenges arise when integrating diffusion models into multimodal frameworks, particularly when interfacing with large language models (LLMs) (Chiang et al., 2023; Touvron et al., 2023). The representations in diffusion models are inconsistent with the embeddings used by LLMs, complicating their seamless integration. This discrepancy might hinder visual generation tasks that require direct collaboration between vision and language models in the future. As a result, these shortcomings necessitate the exploration of more unified approaches to controllable generative modeling.

On the other hand, autoregressive models, drawing inspiration from autoregressive language models (Radford et al., 2019; Brown, 2020), offer a compelling alternative for visual generation tasks. These approaches model image generation as a sequence prediction problem, which align with the representation used in LLMs and offer lower computational costs compared to diffusion models. By employing intricate and scalable designs, recent autoregressive models (Sun et al., 2024; Tian et al., 2024) have demonstrated generative capabilities comparable to those of diffusion models. However, existing autoregressive models have not yet fully explored the potential of controllable visual generation. In an earlier attempt, IQ-VAE (Zhan et al., 2022) introduced condition patches as prefix tokens to generate subsequent image patches. This approach results in overly long sequences, which significantly reduces efficiency. More recently, ControlVAR (Li et al., 2024) models conditions and images simultaneously to guide the generation process. However, it restricts the ability to effectively utilize pre-trained models, thereby increasing the training resources needed and reducing flexibility and adaptability. These inefficiencies underscore the necessity for more versatile and streamlined methods for controllable autoregressive generation.

To address the challenges mentioned above, we propose **C**ontrollable **A**uto**R**egressive Modeling (**CAR**), a novel, end-to-end, plug-and-play framework designed to facilitate controllable visual autoregressive generation by leveraging pre-trained models. The core design of our framework involves integrating multi-scale latent variable modeling, where the control representation is progressively refined and injected into each step of a pre-trained autoregressive model. Specifically, we employ the "next-scale prediction" autoregressive model VAR (Tian et al., 2024) as our pre-trained base, and we freeze its weights to maintain its strong generative capabilities. Inspired by Control-Net (Zhang et al., 2023), we have also designed a parallel control branch to autoregressively model multi-scale control representation, which utilizes both the input condition signal and the embedding from the pre-trained base model. The prediction of each scale's image token map depends on the previous image tokens and the extracted control information. Through this approach, our CAR framework successfully captures multi-scale control representations and injects them into the frozen base model, ensuring that the generated image adheres to the specified visual conditions.

Contributions of this work can be summarized as follows:

1. To the best of our knowledge, our proposed **CAR** is the first flexible, efficient and plug-and-play controllable framework designed for the family of autoregressive models. We hope that our work will contribute to accelerating the development of this field.

2. CAR builds on pre-trained autoregressive models, not only preserving the original generative capabilities but also enabling controlled generation with limited resources—using less

than 10% of the data required for pre-training. We design a general framework to capture multi-scale control representations, which are robust and can be seamlessly integrated into the pre-trained base models.

3. Extensive experiments demonstrate that our CAR achieves precise fine-grained visual control across various condition signals. CAR effectively learns the semantics of these conditions, enabling robust generalization even to unseen categories outside the training set.

## 2 RELATED WORK

**Diffusion Models** Diffusion models have attracted significant attention for their ability to generate high-fidelity images through iterative noise reduction processes. These models operate by progressively transforming Gaussian noise into a data distribution, with each step in the Markov chain refining the image (Sohl-Dickstein et al., 2015; Song et al., 2020). The introduction of the Denoising Diffusion Probabilistic Model (DDPM) (Ho et al., 2020) marked a breakthrough, achieving state-of-the-art results in image synthesis. Following this, several approaches have aimed to improve the efficiency and quality of diffusion models (Nichol & Dhariwal, 2021; Rombach et al., 2022; Watson & Johnson, 2023). In the past two years, diffusion models have nearly become the de facto approach in the realm of text-to-image and text-to-video generation (Saharia et al., 2022; Singer et al., 2022; Peebles & Xie, 2023; Podell et al., 2023; Dai et al., 2023; Blattmann et al., 2023a;b; Esser et al., 2023; 2024). More recently, some works have increasingly focused on integrating diffusion models into multimodal tasks (Nichol et al., 2021; Lu et al., 2022; 2024; Xie et al., 2024; Zhou et al., 2024).

**Autoregressive Models** Autoregressive models have emerged as a scalable alternative to diffusion models in generative tasks, offering a more efficient architecture for image synthesis. Inspired by the success of autoregressive models in language tasks, such as GPT (Radford et al., 2019; Brown, 2020), their visual counterparts like DALL-E (Ramesh et al., 2021) model image generation as a sequence prediction problem. This paradigm shift allows autoregressive models to generate high-quality images while circumventing the iterative nature of diffusion models, thereby reducing computational overhead. A number of excellent works adhering to this paradigm have emerged (Ge et al., 2023; Ma et al., 2024; Lu et al., 2024; 2022; Tian et al., 2024; Team, 2024; Chern et al., 2024; Liu et al., 2024; Dong et al., 2023; Ge et al., 2024).

One of the major developments in this field is the application of discrete latent spaces, introduced by VQ-VAE (Van Den Oord et al., 2017) and VQ-GAN (Esser et al., 2021), enabling efficient encoding and decoding of image data. Subsequent works have further enhanced the representational capacity of discrete visual encoders (Yu et al., 2023a;b; Luo et al., 2024). More recently, VAR (Tian et al., 2024) provides a scaling-up modeling approach for discrete latent spaces, significantly enhancing generation. Nonetheless, while these models exhibit better efficiency and comparable generation quality to diffusion models, they still lack sophisticated controllable generation mechanisms. This limitation restricts their applicability in tasks requiring user-driven or signal-driven generation. Approaches such as ControlVAR (Li et al., 2024) have made some progress; however, they remain inflexible and fail to fully exploit pre-trained models, often necessitating fine-tuning.

**Controllable Generation** Controllable generation, where the model is guided by various conditions during the generative process, has been an active area of research. Early works focused on conditional GANs (Mirza & Osindero, 2014) and VAEs (Kingma & Welling, 2013), where control was imposed through explicit conditioning variables such as class labels. However, the challenge of maintaining both high-quality generation and precise control persists across different generative frameworks. Diffusion-based methods like ControlNet (Zhang et al., 2023) and T2I-Adapter (Mou et al., 2024) have incorporated external control signals, such as pose or sketch, to achieve detailed manipulation of generated content. In contrast, controllable generation methods for autoregressive models, especially efficient ones similar to ControlNet or T2I-Adapter in the diffusion context, have not been fully explored. Actually, prior to our work, it was unknown whether similar capabilities could be achieved with purely autoregressive models. While methods such as IQ-VAE (Zhan et al., 2022) and ControlVAR (Li et al., 2024) allow for fine-grained control over the visual autoregressive generation process by integrating conditional tokens or patches, they cannot flexibly leverage pre-trained models, and increase computational complexity. Therefore, this paper aims to develop a more efficient and flexible controllable framework for autoregressive visual generation.

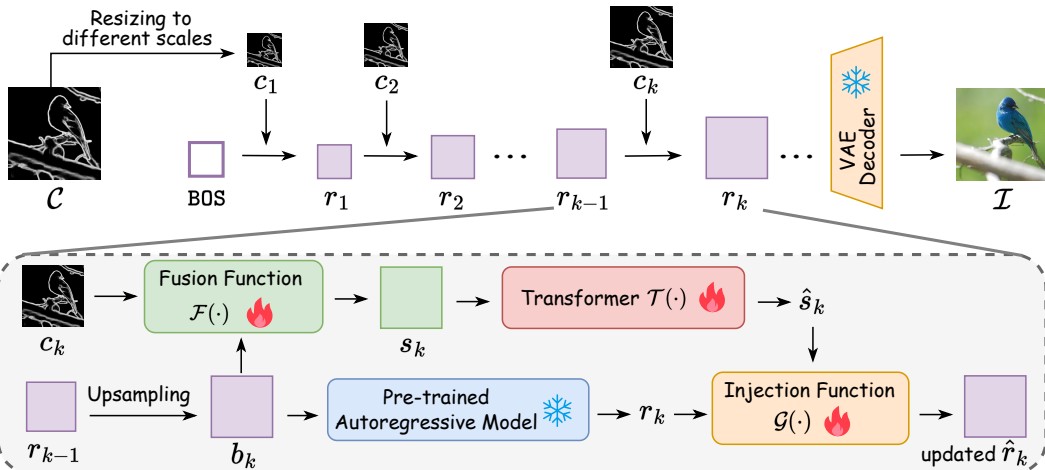

Figure 2: Overview of the proposed **C**ontrollable **A**uto**R**egressive Modeling (**CAR**) framework. CAR integrates multi-scale latent variable modeling, where control representation is progressively refined and injected into the generation process of a pre-trained autoregressive model. Previous image tokens are accumulated and upsampled to form $b_k$, which serves as the input token for scale $k$. Each scale's token map $r_k$ is predicted based on previous tokens and the corresponding control input $c_k$, ensuring that the generated image $\mathcal{I}$ adheres to the specified visual conditions $\mathcal{C}$.

## 3 METHODOLOGY

We propose **C**ontrollable **A**uto**R**egressive Modeling (**CAR**) to explore the potential of autoregressive models in handling controllable image generation task. We define the task as follows: given a conditional control image $\mathcal{C} \in \mathbb{R}^{3 \times H \times W}$, where $H$ and $W$ represent the height and width, our goal is to generate a controllable image $\mathcal{I} \in \mathbb{R}^{3 \times H \times W}$ that aligns with the specified visual conditions. The overall objective can be formulated as modeling the conditional distribution $p(\mathcal{I} \mid \mathcal{C})$.

In Section 3.1, we introduce the preliminary foundational concepts of the "next-scale prediction" paradigm in visual autoregressive modeling. Following this, in Section 3.2, we explain how our proposed CAR framework controls visual generation through multi-scale latent variable modeling. By applying Bayesian inference, we identify that the learning objective of CAR is to obtain a robust control representation. Finally, in Section 3.3, we thoroughly discuss the control representation expression and the network optimization strategy.

### 3.1 PRELIMINARY KNOWLEDGE OF AUTOREGRESSIVE MODELING

Traditional autoregressive models Van Den Oord et al. (2017); Esser et al. (2021) use a "next-token prediction" approach, tokenizing and flattening images into sequences $(x_1, x_2, \ldots, x_T)$. Here, $T$ is the product of the height and width of the image feature map. Each token $x_t$ is predicted based on the preceding tokens $(x_1, x_2, \ldots, x_{t-1})$. The final token sequence is quantized and decoded to produce images.

However, a recent study (Tian et al., 2024) notes that this paradigm can lead to mathematical inconsistencies and structural degradation, which is less optimal for generating highly-structured images. To resolve this, it introduces a novel visual autoregressive modeling paradigm (VAR), shifting from "next-token prediction" to "next-scale prediction". In VAR, each unit predicts an entire token map at a different scale. Starting with a $1 \times 1$ token map $r_1$, VAR predicts a sequence of multi-scale token maps $(r_2, \ldots, r_K)$, increasing in resolution. The generation process is expressed as:

$$p(r_1, r_2, \ldots, r_K) = \prod_{k=1}^{K} p(r_k \mid r_1, r_2, \ldots, r_{k-1}), \tag{1}$$

where $r_k \in [V]^{h_k \times w_k}$ represents the token map at scale $k$, with dimensions $h_k$ and $w_k$, conditioned on previous maps $(r_1, r_2, \ldots, r_{k-1})$. Each token in $r_k$ is an index from the VQVAE codebook $V$, which is trained through multi-scale quantization and shared across scales.

## 3.2 CONTROLLABLE VISUAL AUTOREGRESSIVE MODELING

As illustrated in Figure 2, our proposed CAR framework addresses controllable image generation by modeling the conditional distribution $p(\mathcal{I} \mid \mathcal{C})$. The objective is to maximize the likelihood $p(\mathcal{I} \mid \mathcal{C})$, ensuring that the generated image $I$ conforms to the visual conditions specified by $\mathcal{C}$.

Following the "next-scale prediction" paradigm of VAR, the CAR model adopts a multi-scale latent variable framework, where latent variables (token maps) at each scale capture image structures at progressively higher resolutions. The control information provides additional observations to aid in inferring the latent variables at each scale.

**Multi-scale Conditional Probability Modeling**  Suppose the total number of scales in the latent framework is $K$, our CAR model generates an image $\mathcal{I}$ in a multi-scale fashion by factorizing the conditional distribution $p(\mathcal{I} \mid \mathcal{C})$ as a product of conditional probabilities at each scale:

$$
\begin{aligned}
p(\mathcal{I} \mid \mathcal{C}) &= p(r_1, r_2, \ldots, r_K \mid c_1, c_2, \ldots, c_K) \\
&= \prod_{k=1}^{K} p(r_k \mid (c_1, r_1), (c_2, r_2), \ldots, (c_{k-1}, r_{k-1}), c_k) \\
&= \prod_{k=1}^{K} p(r_k \mid \{(r_i, c_i)\}_{i=1}^{k-1}, c_k),
\end{aligned}
\tag{2}
$$

where $r_k \in \mathbb{R}^{h_k \times w_k}$ is the image token map at scale $k$, and $c_k \in \mathbb{R}^{h_k \times w_k}$ is the corresponding control map derived from the control image $\mathcal{C}$. Each token map $r_k$ is generated conditioned on the previous token maps $(r_1, r_2, \ldots, r_{k-1})$ and the control information $(c_1, c_2, \ldots, c_k)$. This multi-scale conditional modeling ensures that the control information at each scale guides the generation process in a recursive and hierarchical manner, progressively refining the latent representations of the image token maps across scales.

**Posterior Approximation**  In the CAR framework, as formulated in Equation 2, the previous scales' image and control token maps, $\{r_i, c_i\}_{i=1}^{k-1}$, along with the current scale's control token map $c_k$, serve as a posterior approximation for the current scale's image token map. This means that $r_k$ is generated by leveraging the information from this posterior approximation. From a Bayesian perspective, the goal of the CAR model is to approximate the posterior distribution of the image tokens given the control information at each scale:

$$
p(r_k \mid \{(r_i, c_i)\}_{i=1}^{k-1}, c_k) \propto p(r_k \mid \{r_i, c_i\}_{i=1}^{k-1}) p(c_k \mid r_k),
\tag{3}
$$

where $p(r_k \mid \{r_i, c_i\}_{i=1}^{k-1})$ represents the autoregressive prior learned across scales from previous token maps, and $p(c_k \mid r_k)$ captures the likelihood of observing the control token map $c_k$ given the current image token map $r_k$.

Based on the above Bayesian inference, we can clearly identify that the learning objective of CAR is to optimize $c_k$ so that this control representation aligns with the image representation $r_k$. This objective can be learned through a neural network, supervised by the ground truth $r_k$ from the provided image dataset, allowing the network to progressively approximate the posterior distribution.

## 3.3 CONTROL REPRESENTATION AND OPTIMIZATION

**Control Representation Expression**  VAR accumulates image tokens $\{r_i\}_{i=1}^{k-1}$ from all previous scales, interpolates them to match the resolution of $h_k \times w_k$, and forms the input for inference at scale $k$, denoted as $b_k$. Then in our CAR framework, at each scale $k$, the control information is injected by fusing the input image token map $b_k \in \mathbb{R}^{h_k \times w_k \times d}$ and the control map $c_k \in \mathbb{R}^{h_k \times w_k \times d}$ to form a combined representation:

$$
s_k = \mathcal{F}(b_k, c_k; \theta_{\mathcal{F}}),
\tag{4}
$$

where $\mathcal{F}(\cdot)$ is a fusion function parameterized by $\theta_{\mathcal{F}}$. This fusion mechanism ensures that $s_k$ encapsulates both generated image features and the control conditions. By utilizing $s_k$ to predict $r_k$, the control information is incorporated into the generation process, ensuring that the generated token map adheres to the control conditions $c_k$, providing fine-grained guidance.

To ensure effective extraction of control representation and integration of control information, the fused representation $s_k$ is vectorized and transformed via a series of Transformer layers, yielding a refined conditional prior that guides the image generation at each scale $k$. Formally, $s_k$ is transformed into a vectorized representation $\hat{s}_k$ through a learned mapping $\mathcal{T}(\cdot)$ as follows:

$$\hat{s}_k = \mathcal{T}(s_k; \theta_{\mathcal{T}}), \tag{5}$$

where $\mathcal{T}(\cdot)$ is parameterized by the CAR's Transformer $\theta_{\mathcal{T}}$, which is designed as a parallel branch alongside the VAR's Transformer. The blocks in $\mathcal{T}(\cdot)$ perform self-attention on the vectorized control representations, extracting relevant condition priors by modeling dependencies within the control information.

Once the refined conditional prior $\hat{s}_k$ is extracted, it is injected into the image token map $r_k$, which is predicted by the pre-trained VAR and represents the latent image features at scale $k$. This injection is achieved through a injection function $\mathcal{G}(\cdot)$ parameterized by $\theta_{\mathcal{G}}$, which combines $\hat{s}_k$ and $r_k$ to ensure that the control information modulates the generated image tokens:

$$\hat{r}_k = \mathcal{G}(\hat{s}_k, r_k; \theta_{\mathcal{G}}), \tag{6}$$

where $\hat{r}_k$ represents the updated token map incorporating the control information. This mechanism enables the model to progressively condition the generation process on multi-scale control information, thereby producing images that are coherent across scales and adhere to the external visual condition specified by $\mathcal{C}$.

**Network Optimization** To align the generated image $\mathcal{I}$ with the control conditions $\mathcal{C}$, we minimize the Kullback-Leibler (KL) divergence (Kullback & Leibler, 1951) between the model's conditional distribution $p(\mathcal{I} \mid \mathcal{C}; \theta_{\text{car}})$ and the true data distribution $p_{\text{data}}(\mathcal{I} \mid \mathcal{C})$:

$$\mathcal{L}_{\text{KL}} = \mathbb{E}_{\mathcal{I}, \mathcal{C} \sim p_{\text{data}}} \left[ \log p_{\text{data}}(\mathcal{I} \mid \mathcal{C}) - \log p(\mathcal{I} \mid \mathcal{C}; \theta_{\text{car}}) \right], \tag{7}$$

where $\theta_{\text{car}}$ represents the learnable parameters in our CAR framework, specifically $\{\theta_{\mathcal{F}}, \theta_{\mathcal{T}}, \theta_{\mathcal{G}}\}$. Since $p_{\text{data}}(\mathcal{I} \mid \mathcal{C})$ is constant with respect to $\theta_{\text{car}}$, minimizing $\mathcal{L}_{\text{KL}}$ is equivalent to maximizing the log-likelihood $\log p(\mathcal{I} \mid \mathcal{C}; \theta_{\text{car}})$.

Using the fused representations $s_k$, the conditional distribution $p(\mathcal{I} \mid \mathcal{C}; \theta_{\text{car}})$ can be factorized as:

$$\log p(\mathcal{I} \mid \mathcal{C}; \theta_{\text{car}}) = \sum_{k=1}^{K} \log p(\hat{r}_k \mid s_k; \theta_{\text{car}}). \tag{8}$$

Maximizing this log-likelihood during training ensures that the generated token maps $\hat{r}_k$ closely match the ground truth $r_k$, remaining consistent with both previous tokens and the control conditions encapsulated in $s_k$. This process facilitates the learning of more effective control representations, as outlined in the posterior approximation in Equation 3, ensuring that the generated images adhere to the control conditions $\mathcal{C}$ while preserving the original generative capabilities of pre-trained VAR.

# 4 EXPERIMENTS

## 4.1 EXPERIMENTAL SETUPS

**Model Architecture Design** We employ the pre-trained VAR (Tian et al., 2024) as the base model, freezing it during training to preserve its generative ability and reduce training costs. For the learnable modules $\{\theta_{\mathcal{F}}, \theta_{\mathcal{T}}, \theta_{\mathcal{G}}\}$, we experiment with various choices and empirically select the optimal design choices. For the fusion function $\mathcal{F}(\cdot)$, we use a convolutional encoder to extract semantic features from the control input $c_k$, and add them to the base model input $b_k$. For $\mathcal{T}(\cdot)$, we design a series of GPT-2-style Transformer (Radford et al., 2019) blocks, with the depth being half of that of the pre-trained base model. For $\mathcal{G}(\cdot)$, we inject $\hat{s}_k$ into the base model output $r_k$ through concatenation, which is followed by a LayerNorm (Ba, 2016) to normalize the distribution of the two domain features, and a linear transformation to adjust the channel dimension.

Table 1: Comparison with previous controllable generation approaches on ImageNet. Our CAR surpasses these works by delivering higher image quality while being much more efficient in inference.

| Methods | Canny Edge | | Depth Map | | Normal Map | | HED Map | | Sketch | | Time (s) ↓ |
| | FID ↓ | IS ↑ | FID ↓ | IS ↑ | FID ↓ | IS ↑ | FID ↓ | IS ↑ | FID ↓ | IS ↑ | |
| --- | --- | --- | --- | --- | --- | --- | --- | --- | --- | --- | --- |
| T2I-Adapter | 10.2 | 156.6 | 9.9 | 133.6 | 9.5 | 142.8 | 9.3 | 141.6 | 16.2 | 156.0 | 2.3 |
| ControlNet | 11.6 | **172.7** | 9.2 | 150.3 | 8.9 | 155.3 | 8.6 | 150.3 | 15.3 | **162.5** | 1.7 |
| **CAR (Ours)** | **8.3** | 167.3 | **6.9** | **178.6** | **6.6** | **175.9** | **5.6** | **182.2** | **10.2** | 161.6 | **0.3** |

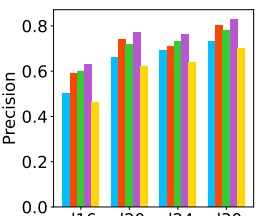 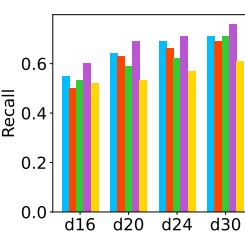 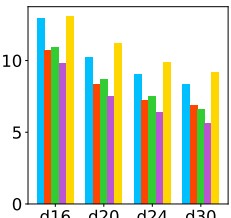 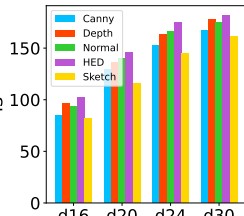

Figure 3: Scaling laws of our CAR model. It can be observed that as the model depth increases, the four image quality metrics improve across the five conditions.

**Dataset**   We conduct experiments on the ImageNet (Russakovsky et al., 2015) dataset. First, we pseudo-label five conditions for the training set: Canny edge (Canny, 1986), Depth map (Ranftl et al., 2020), Normal map (Vasiljevic et al., 2019), HED map (Xie & Tu, 2015), and Sketch (Su et al., 2021), allowing CAR to be trained separately on different conditional controls. We randomly select 100 categories from the total 1000 for training CAR, and evaluate it on the remaining 900 unseen categories to assess its generalizable controllability.

**Evaluation Metrics**   We utilize Fréchet Inception Distance (FID) (Heusel et al., 2017), Inception Score (IS) (Salimans et al., 2016), Precision and Recall (Kynkäänniemi et al., 2019) metrics to assess the quality of the generated results. We also compare the inference speed with existing controllable generation methods, such as ControlNet (Zhang et al., 2023) and T2I-Adapter (Mou et al., 2024).

**Training Details**   We set the depth of the pre-trained VAR to 16, 20, 24, or 30, and initialize the control Transformer $\mathcal{T}(\cdot)$ using the weights from the first half of the VAR to accelerate convergence. The CAR model is trained for 100 epochs with the Adam optimizer on 8 NVIDIA V100 GPUs, and the inference speed is evaluated on a single NVIDIA 4090 GPU.

### 4.2 QUANTITATIVE EVALUATION

**Comparison with Previous Methods**   We compare our CAR model with two classic controllable generation baselines, ControlNet and T2I-Adapter. For a fair comparison, we retrained both models on the ImageNet dataset and trained each model separately on all five condition annotations. As shown in Table 1, our CAR demonstrates significant improvements, with FID reductions of 3.3, 2.3, 2.3, 3.0, and 5.1 on Canny, Depth, Normal, HED, and Sketch, respectively, compared to ControlNet. Similar improvements are observed in the IS metric. We attribute these gains to recent advancements in autoregressive models, which have surpassed diffusion models in image generation by progressively scaling up the resolution during generation. In addition to image quality, we also compare inference speed, with our CAR running over five times faster than both ControlNet and T2I-Adapter, further highlighting the efficiency advantage of CAR in practical applications. Overall, these promising quantitative results indicate that CAR can serve as a more efficient and scalable controllable generative paradigm than diffusion-based models like ControlNet.

As for different types of conditions, it is worth noting that HED Maps, Depth Maps, and Normal Maps demonstrate relatively superior metrics, likely due to the clarity of input conditions and well-defined objectives. These factors provide the model with more precise guidance, enhancing the generation of high-quality images. In contrast, the Sketch condition is often simplistic, consisting

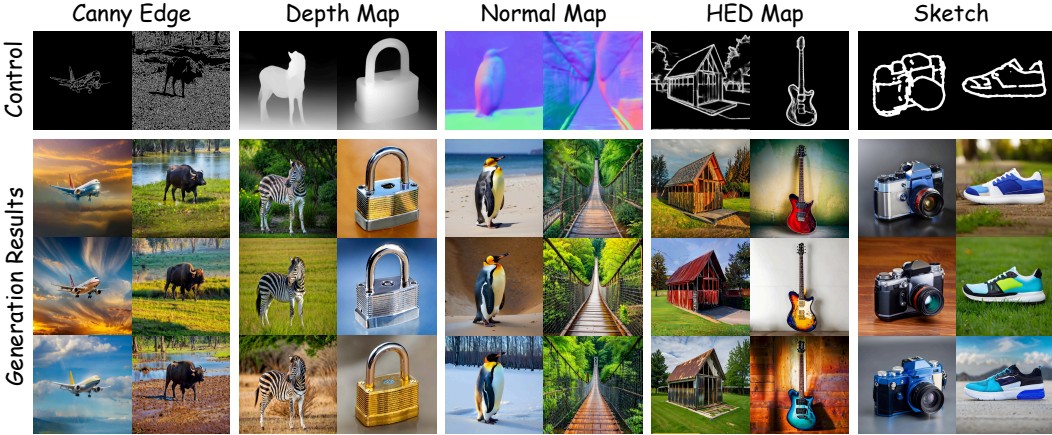

Figure 4: Results are presented for five distinct conditions, where the top row shows the input conditions, and the following rows display the generated images. These categories are excluded from the training set, demonstrating that the CAR learns the general semantics from the input conditions.

of basic outlines with fewer visual details compared to other conditions, making it less controllable and leading the model to generate more freely. This may result in fluctuations in image quality.

**Scaling Laws**    We assess the image quality of our CAR model as its depth increases. As illustrated in Figure 3, with increasing model depth, the CAR produces higher-quality images across five different conditions, demonstrating a lower FID metric alongside elevated IS, Precision, and Recall scores, which align with the scaling laws of autoregressive generative modeling (Kaplan et al., 2020; Henighan et al., 2020; Hoffmann et al., 2022). The most high metrics are observed in HED Maps, Depth Maps, and Normal Maps, and Canny Edge and Sketch are relatively low, which is consistent with the observation in Table 1.

**User Studies**    We conduct the user studies with 30 participants to evaluate the generation performance of our CAR in comparison with previous methods, ControlNet and T2I-Adapter. For each of the five types of conditions, we input 30 condition images and generate corresponding results for each method, producing 150 results per method. For each conditional input, participants are required to choose the best one based on three criteria: 1) image quality, 2) condition fidelity, and 3) image diversity. As demonstrated in Table 2, our CAR

Table 2: User studies are conducted to evaluate three controllable generative approaches based on three criteria: **1) IQ**: image quality, **2) CF**: condition fidelity, and **3) ID**: image diversity.

| Methods | IQ ↑ | CF ↑ | ID ↑ |
|---|---|---|---|
| T2I-Adapter | 23% | 27% | 19% |
| ControlNet | 26% | 31% | 36% |
| **CAR (Ours)** | **51%** | **42%** | **45%** |

outperforms ControlNet and T2I-Adapter in all three aspects, demonstrating the effectiveness of proposed Controllable Autoregressive Modeling.

### 4.3    QUALITATIVE RESULTS

**Overall Controllability and Image Quality**    Figure 4 qualitatively demonstrates that our CAR model generates high-quality and diverse results based on the given conditional controls. The visual details of various conditional inputs are effectively reflected in the generated images, ensuring a strong alignment between the images and their corresponding conditions. Notably, the categories shown are not among the 100 categories used during training, yet CAR still achieves precise control over these unseen categories, demonstrating that our CAR learns the general semantic information from the given conditional controls, rather than overfitting to the training set. This advantage highlights the cross-category generalization and robust controllability of our CAR framework.

Table 3: Comparisons of IS metrics for different function choices, including $\mathcal{F}(\cdot)$, $\mathcal{T}(\cdot)$, and $\mathcal{G}(\cdot)$.

| | Settings | Canny | Depth | Normal | HED | Sketch |
|---|---|---|---|---|---|---|
| $\mathcal{F}(\cdot)$ | Pre-trained VQ-VAE Encoder | 131.3 | 139.2 | 141.0 | 149.7 | 123.5 |
| | **Learnable Convolutional Encoder** | **166.2** | **177.9** | **173.0** | **181.9** | **159.3** |
| $\mathcal{T}(\cdot)$ | Convolutional Network | 145.7 | 156.8 | 153.2 | 159.6 | 142.6 |
| | **Transformer Network** | **166.2** | **177.9** | **173.0** | **181.9** | **159.3** |
| $\mathcal{G}(\cdot)$ | Zero Convolution & Add | 161.9 | 170.3 | 168.7 | 173.5 | 153.8 |
| | Cross Normalization & Add | 160.7 | 170.1 | 169.3 | 173.1 | 154.2 |
| | **Concat & LayerNorm & Linear** | **166.2** | **177.9** | **173.0** | **181.9** | **159.3** |

**Analysis of Data Distribution** We analyze the controllability of CAR from the perspective of data distribution. Specifically, HED Maps are used as a type of condition to guide the image generation process, with this condition extracted from ground truth images. An uncontrollable vanilla autoregressive model (Tian et al., 2024) is employed to generate comparison samples. We apply t-SNE (Van der Maaten & Hinton, 2008) to visualize the first two principal components of the embedding features from all generated images. These embedding features are extracted using the backbone of the HED Map extraction method (Xie & Tu, 2015).

As illustrated in Figure 5, there is a significant misalignment between the generation distribution of the vanilla autoregressive model and the ground truth, as the vanilla model lacks condition control information. In contrast,

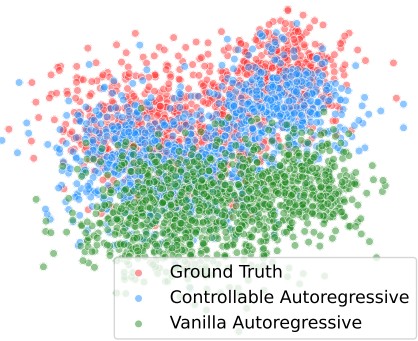

Figure 5: T-SNE visualization of the distribution of generation results from our CAR model and the vanilla model.

the distribution of the CAR model's generated results closely aligns with the ground truth, demonstrating that our samples accurately capture the visual details of the HED Map, bringing the HED embedding features closer to the ground truth. This highlights that our CAR model enhances both the controllability and accuracy of generated results $\mathcal{I}$ based on the provided condition control $\mathcal{C}$.

### 4.4 Ablation Studies

We conduct ablation studies on the ImageNet validation set to explore the different function choices for each component in the CAR framework, including $\mathcal{F}(\cdot)$, $\mathcal{T}(\cdot)$, and $\mathcal{G}(\cdot)$.

**Different Function Choices for $\mathcal{F}(\cdot)$** We explore the impact of different methods for introducing conditional control $c_k$ to form $s_k$ in $\mathcal{F}(\cdot)$. Specifically, we compare two strategies: **1)** using the pre-trained VQ-VAE encoder from the VAR model to directly map the condition image to token maps at various scales; **2)** our approach, which resizes the condition images to different scales at the pixel level and employs a shared, learnable convolutional encoder for control feature extraction. The results are shown in Table 3, where the learnable encoder shows significant improvements in IS scores, indicating enhanced image quality. We hypothesize that the pre-trained VQ-VAE encoder, being designed for image reconstruction, may not effectively capture image semantics, making it less suitable for extracting control semantics. Similar visualization results are illustrated in Figure 6, where using the VQ-VAE encoder results in image distortion and poor quality.

**Different Function Choices for $\mathcal{T}(\cdot)$** We design an encoder for $\mathcal{T}(\cdot)$ to extract accurate and effective control representations $\hat{s}_k$. Specifically, we compare two architectures: **1)** a simple convolutional network, and **2)** a GPT-2-style Transformer. As shown in Table 3 and Figure 6, the Transformer shows significantly higher image quality compared to the simple convolutional network baseline, due to its powerful representation ability. Meanwhile, the Transformer-based en-

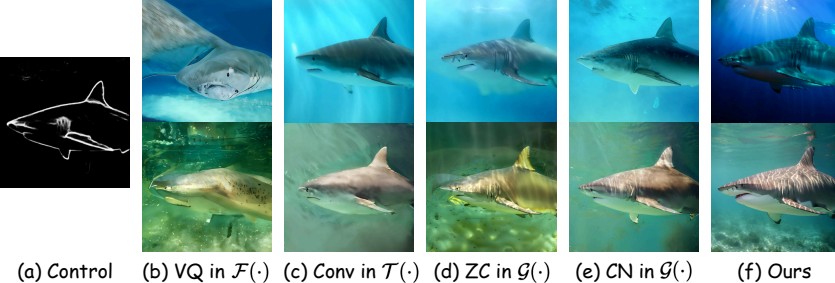

(a) Control  (b) VQ in $\mathcal{F}(\cdot)$  (c) Conv in $\mathcal{T}(\cdot)$  (d) ZC in $\mathcal{G}(\cdot)$  (e) CN in $\mathcal{G}(\cdot)$  (f) Ours

Figure 6: Comparison of different function choices given **(a)** control input: **(b)** using a pre-trained VQ-VAE encoder in $\mathcal{F}(\cdot)$, **(c)** designing a convolutional network in $\mathcal{T}(\cdot)$, **(d)** applying zero convolution in $\mathcal{G}(\cdot)$, **(e)** using cross normalization in $\mathcal{G}(\cdot)$, and **(f)** our final architecture.

coder aligns with the architecture of the pre-trained autoregressive model, which may result in a closer distribution, enhancing the subsequent injection process.

**Different Function Choices for $\mathcal{G}(\cdot)$**   We compare different injection functions $\mathcal{G}(\cdot)$, where the control representation $\hat{s}_k$ is injected into the image representation $r_k$ of a pre-trained autoregressive model to update the image representation $\hat{r}_k$. Specifically, we compare three techniques: **1)** applying zero convolutions (Zhang et al., 2023) to the control representation, followed by the addition of control and image features; **2)** applying cross normalization (Peng et al., 2024), which normalizes the control representation using the mean and variance of the image representation, then adds these two features; **3)** our method, which concatenates the two representations, applies a learnable LayerNorm to normalize the distributions, followed by a linear transformation to adjust the channel dimension.

As shown in Table 3, adding the image and control features leads to a decrease in the IS metric, regardless of whether zero convolution and cross normalization are applied before the addition. This indicates that these operations result in a reduction in image quality compared to our approach. The generation results in Figure 6 also demonstrate that these two baselines perform worse than our approach in terms of image quality and naturalness. We attribute this to the incompatibility of two different domain representations. Although cross normalization tries to align the distribution across the domain gap, such an operation is insufficient. Therefore, concatenating the two representations, followed by LayerNorm, more effectively harmonizes the conditional and backbone features, addressing discrepancies in data distribution.

## 5   CONCLUSION

In this paper, we propose **C**ontrollable **A**uto**R**egressive Modeling (**CAR**), which establishes a novel paradigm for controlling VAR generation. CAR captures robust multi-scale control representations, which can be seamlessly integrated into pre-trained autoregressive models. Our experimental results demonstrate that CAR outperforms existing methods in both controllability and image quality, while also reducing the required computational costs. CAR represents a significant step forward in autoregressive visual generation, offering a flexible, efficient, and scalable solution for various controllable generation tasks.

**Discussion and Future Works**   While the proposed **CAR** framework demonstrates advancements in controllable visual generation, it still faces certain limitations inherent in the underlying **VAR** model. Specifically, the reliance on sequential token prediction can sometimes limit the model's efficiency, especially when dealing with long image sequences or when requiring precise fine-grained control at high resolutions. The multi-scale injection mechanism used in CAR could also be extended to explore alternative injection strategies, such as attention-based or adaptive injection, to further enhance control precision. Additionally, although the current design excels at injecting control signals in a recursive manner, extending the framework to handle more complex tasks, such as video generation, remains an open challenge for future work.

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
