# OpenReview forum: "CAR: Controllable Autoregressive Modeling for Visual Generation"
_ICLR.cc/2025/Conference — ICLR 2025 Conference Withdrawn Submission_

### Official Review · Reviewer_nq9U · 2024-10-31

**Soundness:** 3
**Presentation:** 2
**Contribution:** 2
**Rating:** 3
**Confidence:** 4

**Summary:**

This paper proposed a plug and play method for scale-based autoregressive conditional image generation.
The authors trains a side network to fuse and inject the conditional information into a pre-trained VAR.
The proposed method was trained on 100 categories and shows generalizable controllability on other classes.
The results shows superiority over T2I-Adapter and ControlNet.

**Strengths:**

* The paper is easy to follow.
* The proposed method maintains the pre-trained VAR network unchanged, and is trainable with only 8 V100.
* Detailed ablation study over the network design is provided.

**Weaknesses:**

* Some important information is missing from the paper and there is no appendix to explain those points. See questions.
* The problem solved in this paper is same as in ControlVAR, yet it has not compared with this baseline method. In fact, the results from ControlVAR is better than this paper.
* Derivation of eq 3 is not obvious and not detailed information.

**Questions:**

1. What are the number of parameters of the proposed fusion function, transformer, and injection function?

2. What are the flops of the proposed method compared to ControlVAR. From current results, there is no demonstration of efficiency as claimed in the paper. While the authors showed in Table 1 that the inference time is faster than T2I-Adapter and ControlNet, I believe this mainly comes from the benefits of VAR over diffusion models.

3. What is the scalability of the proposed method over the number of data. Why 100 categorizes are chosen? Does the method work for less classes? Does the method work for more classes?

---

> ### Author Response · Authors · 2024-11-15
> **We would be more grateful if the feedback you provided was about the novelty and contributions of our work.**
>
> Thank you for your detailed feedback,  but what do you think about the novelty and contribution of our work?
> We appreciate the thorough review and understand that addressing minor issues is part of the peer review process. However, we feel that the numerical and technical details you raised, while important, do not fully capture the core idea of our work. We believe that evaluating whether our work provides valuable insights and advancements to the community is equally important.
>
> Despite our disagreement that these detail issues are sufficient reason for the low rating score, we still find your feedback valuable for improving the quality of our paper. Here are our specific responses to your questions:
>
> **More comparison with ControlVAR**: We will provide a detailed comparison with ControlVAR in the revised version. However, it is important to note that we do not consider ControlVAR to be superior to our method, as was suggested. ControlVAR requires doubling the number of output tokens, which significantly increases the computational cost for both training and inference. Additionally, it necessitates full training on ImageNet. In contrast, our method only requires training 50% of the original parameters, 10% of the original training data, and 50% of the original training epochs, resulting in a total training computational cost of just 2.5%. Actually, if we do not consider generalization, even 1% of the training data is sufficient. This makes our method far more efficient compared to ControlVAR.
>
> **Details of Eq3 and Model Architecture**: We will provide more detailed information about Eq3, the proposed fusion function, transformer, and injection function, including their parameter counts. However, we want to emphasize that these components add minimal additional parameters and are minor issues. Our focus is on the overall effectiveness and efficiency of the control mechanism.
>
> **Number of Categories**: The choice of 100 categories was somewhat arbitrary due to our limited computational resources (using 8 V100 GPUs). After testing with 100 categories, we observed satisfactory control effects and quantitative metrics, so we did not explore more categories. However, our method is flexible and can handle more or fewer categories. For instance, we have tested with 10 categories and achieved good visualization results.
>
> We hope these clarifications help to better convey the significance and value of our work. We are committed to improving our manuscript and ensuring that our contributions are clearly communicated to the community.

---

### Official Review · Reviewer_mVjS · 2024-11-02

**Soundness:** 3
**Presentation:** 1
**Contribution:** 2
**Rating:** 5
**Confidence:** 4

**Summary:**

This paper proposes CAR designed to adapt pre-trained AR-based image generation models with additional pixel-level controls. A fusion function and a transformer-based adapter are leveraged to inject control information into the original model. The proposed approach demonstrates superior performance compared to retrained ControlNet and T2I-Adapter.

**Strengths:**

- AR-based visual generation is popular.
- The proposed approach was tested with different control types.

**Weaknesses:**

- The "control token", "control map", and "control information" are mixed to describe $c_k$ which is very confusing.
- The model architecture (such as $\mathcal{F}$ and $\mathcal{T}$) is described in the experiment section making it hard to understand the workflow of the proposed approach.
- The proposed approach introduces $\mathcal{T}$ which has 0.5 #param of the original model + several fusion modules $\mathcal{F}$ making the model much larger compared to VAR.
- The author claimed that CAR is the first framework to adapt pre-trained AR-based image generation approaches to additional conditions. However, there exist previous approaches designed for AR models, such as Lumina-MGPT.
- The proposed approach is motivated by improving data and model efficiency. However, it costs 100 epochs on 10% of the ImageNet with a larger model with additional $\mathcal{F}+\mathcal{T}+\mathcal{G}$. The original VAR is just trained with 200-350 epochs.
- Only a few baseline methods (ControlNet and T2I-adapter) are compared. What about the performance compared to ControlVAR?
- The performance of the proposed approach is fair.

**Questions:**

- Only $r_k$ is discussed in Sec 3.1. How to obtain the control map $c_k$? Is it a quantized feature or a resized raw input?
- A comparison of the speed and model size against VAR is missing.
- I am curious about why is the model trained with an adapter-based approach instead of LORA.
- More qualitative results will be helpful in understanding the performance.

I initialed the rating as 3 and will increase it if my concerns can be well addressed.

---

> ### Author Response · Authors · 2024-11-15
> **We believe that we are the first to achieve controllability while maintaining visual AR models‘s inherent capabilities.**
>
> Thank you for raising many detailed questions. We will address them one by one. However, we still believe that these detailed issues do not overshadow our core contribution, which merits a more favorable initial score. Our work is the first to achieve controllability while maintaining the inherent capabilities of AR models, a goal that existing solutions have not been able to accomplish. Below are our specific responses to your concerns:
>
> **Weakness 1 & 2:** We will correct the writing issues regarding the mixing of terms like "control token," "control map," and "control information," as well as the placement of the model architecture description, in future versions. Thank you for pointing this out.
>
> **Weakness 3:** While our method does indeed add approximately 0.5 times the parameters of the original model, this is also the case with the popular ControlNet method used in diffusion models. This trade-off brings stability and rapid convergence during training. We believe there is currently no method that can maintain the model's original capabilities and achieve good controllability with a negligible increase in parameters.
>
> **Weakness 4:** We did not notice the work Lumina-MGPT, which was published in August. However, ICLR submissions's deadline is in September, and according to the reviewer guidelines, papers published on arXiv within the past four months can be considered concurrent work. Therefore, we do not consider this an oversight. Nevertheless, we reviewed this work and found that it is fundamentally different from ours. Lumina-MGPT is a multimodal model fine-tuned on a powerful Chameleon base model, using over 50 million text-image pairs and finally completely altering the pretrained model's parameters. In contrast, our method focuses solely on pure AR visual generation and only trains the control capabilities on top of a frozen pretrained model using merely 0.1 million data points. Comparing the two is neither fair nor meaningful. However, we would like cite this work and explain the differences in our next submission.
>
> **Weakness 5:** We believe the concern about efficiency is misplaced. Our method uses only 50% of the trainable parameters, 10% of the training data, and up to 50% of the training epochs of the original model. When combined, the total computational cost is only 2.5% of the pretraining process. Even with this reduced computational load, we achieved comparable or better control performance without losing generalization. Compared to methods like ControlVAR, which require full training, our method offers significant efficiency advantages and is currently the most efficient approach for controlling AR models.
>
> **Weakness 6:** We will add more comprehensive comparisons with ControlVAR and other relevant methods.
>
> **Question 1:** The control map is obtained by resizing the raw input image and then passing it through several convolutional layers.
>
> **Question 2:** We will supplement the comparison with the original model, noting that the model size increases by 50%, and inference speed is also affected. However, this is not the primary focus of control-oriented work.
>
> **Question 3:** To our knowledge, LoRA (Low-Rank Adaptation) methods are not applicable in the context of introducing control conditions in both diffusion and AR models. LoRA is designed for parameter tuning and cannot introduce external control signals.
>
> **Question 4:** Thank you for the suggestion. We will strive to include more quantitative experimental results.
>
> We hope these clarifications address your concerns.

---

> > ### Comment · Reviewer_mVjS · 2024-11-15
> >
> > I appreciate the author's effort to make the rebuttal. I am increasing the initial rating to 5 to reflect it.
> >
> > I encourage the authors to address the writing and efficiency issues and resubmit.

---

### Official Review · Reviewer_G2iA · 2024-11-04

**Soundness:** 3
**Presentation:** 3
**Contribution:** 3
**Rating:** 6
**Confidence:** 3

**Summary:**

This paper proposes the first control framework for autoregressive generation models, achieving controllability validated under various conditions like Canny, depth, and HED annotations. The results indicate effectiveness, but as I am not an expert in this specific domain, I am inclined to give **borderline acceptance** and look forward to other reviewers' feedback to ensure nothing has been overlooked.

**Strengths:**

1. The paper presents a novel contribution, as CAR is the first framework for controllable autoregressive image generation.
2. The analysis in Section 4.3 clearly shows that controllable autoregressive modeling functions effectively.
3. The presentation of the paper is well done, with clear equations, figures, and tables.

**Weaknesses:**

1. In Section 4.2 (line 643), the authors mention retraining T2I-Adapter and ControlNet but do not provide sufficient details about whether these models were trained with the same parameters and training time as CAR. Additionally, it is worth noting that both methods are based on diffusion models, which seems slightly unconventional. While there may not be directly comparable models, having a stronger baseline would be beneficial.
2. The supplementary material lacks extensive visualizations, with only source code provided. Additional visual results would enhance the paper's comprehensiveness.

**Questions:**

1. A minor question: In Figure 2, when using Canny as a condition (assuming $c$ represents Canny), is $c_1$  obtained by resizing? If so, does the Canny information retain its integrity after downsampling?

---

> ### Author Response · Authors · 2024-11-15
> **CAR is the First Control Method to Retain the Original Capabilities of Visual AR Models**
>
> We appreciate your objective and detailed feedback and are committed to addressing your concerns. Below are our responses to the specific points you raised:
>
> **Weaknesses**:
> **1**.Training Details for Baselines:
> We acknowledge that in Section 4.2 , we did not provide sufficient details about the training settings for T2I-Adapter and ControlNet for fair comparision. We would like to supplement this information to ensure clarity.
> **2**.Supplementary Material:
> While we agree that the code is crucial, we also recognize the importance of visualizations. We will enhance the supplementary material by adding more extensive visual results, Thank you for your suggestion.
>
> **Questions**: Canny Conditioning in Figure 2:
> Yes, c_1 and other images are obtained by resizing. Since the resizing involves scaling down from larger images, there might be some loss of detail, but the overall structure and integrity of the Canny information are retained. This should not significantly affect the prediction of tokens at smaller scales, and our experiments have confirmed the effectiveness of this approach.

---

### Official Review · Reviewer_6frS · 2024-11-05

**Soundness:** 3
**Presentation:** 3
**Contribution:** 2
**Rating:** 3
**Confidence:** 4

**Summary:**

The work proposes controllable autoregressive modeling (CAR), a conditional control module for VAR image generation. Following VAR, CAR adds control to pretrained model with multi-scale latents in a progressive manner. Experiments on multiple conditional generation task show competitive performance of CAR over baselines include T2I-Adapter and ControlNet.

**Strengths:**

1. The motivation and formulation of the work is clear, namely investigating conditional control of AR image generation model.
2. The model show competitive performance on various conditional generation tasks.
3. The paper is clearly written and easy to follow.

**Weaknesses:**

Two major concerns of this work:
1. The model follows VAR which leverages multi-scale latents in generation. This greatly limits the application to broader autoregressive image generation models, where no multi-scale latent is used.
2. The other concern is the computational overhead has not been properly reported. The Transformer module in CAR is built with half of the parameters of original VAR, which can be expensive in training/inference.

**Questions:**

1. Can you authors provide more implementation details of baseline ControlNet and T2I-Adapter?
2. How does the training cost and convergence speed of CAR compared with baseline models?
3. How can CAR be applied to other AR generative models that don't use multi-scale tokens?
4. Also, how does it work if use small-sized CAR for large pretrained VAR model?

---

> ### Author Response · Authors · 2024-11-15
> **Despite your mentioned specific limitations, our method is still a pioneering and significant advancement in the field of controlled generation for visual AR models.**
>
> Thank you for your insightful comments and suggestions, they have helped us to better understand potential areas for improvement in our work. Below, we address the concerns and questions raised in your review:
>
> **Weakness 1:** we acknowledge that our current experiments are limited to VAR models operating under the next-scale paradigm. However, we believe that our methodology could be adapted to other AR models, such as those using the next-token approach in LLamaGen or the masked autoregressive technique in MAR, by incorporating conditions during the decoding process based on the spatial positioning of visual tokens. While this represents a current limitation, we view it more as an opportunity for future exploration and expansion of our work. It is also worth noting that CAR stands out as the first method to achieve controllable generation while preserving the original capabilities of visual AR models, something that, to the best of our knowledge, has not been achieved by other methods like ControlVAR or ControlAR, which typically modify the original model parameters and capabilities. CAR's modular design, allowing for inference with the same base model through simple parameter replacement, underscores its flexibility and adaptability. We will strive to further highlight these aspects in our subsequent revisions.
>
> **Weakness 2:** Concerning the second limitation regarding computational overhead, we appreciate your feedback. It is true that the Transformer module in CAR utilizes approximately half of the parameters of the original VAR model, which suggest higher costs  inference. However, it is important to note that similar techniques, such as ControlNet in diffusion models, employ an equivalent strategy without facing significant criticism regarding computational efficiency. In fact, ControlNet is celebrated for its capacity to deliver effective control outcomes with minimal training data and epochs. Our method achieves comparable results with just half the training parameters of the original model, alongside a mere 10% of the training data and fewer than 50% of the training epochs, all while requiring aprroximately 2.5% of the computational resources needed for the initial pretraining phase. We hope this clarifies any concerns regarding the computational demands of our approach.
>
> To address your questions:
>
> **1**. Regarding the implementation details of the baseline models ControlNet and T2I-Adapter, we agree that providing more context could be beneficial. However, given the established recognition of these models within the industry, we opted to focus our paper on the innovative aspects of CAR. We will consider adding a brief overview of these models in our next revision to offer readers a more comprehensive understanding.
>
> **2**. Comparing the training cost and convergence speed of CAR with baseline models is a valid point. While we agree that such comparisons are valuable, they may not fully capture the unique characteristics of AR models versus diffusion models. The differences observed could largely be attributed to the intrinsic properties of these model types rather than the control mechanisms employed. Nonetheless, we will reflect on how to better contextualize these aspects in our discussion.
>
> **3**. Your question about applying CAR to other AR generative models that do not utilize multi-scale tokens is insightful. Although our current experiments are centered around models with multi-scale latents, we are optimistic that the core principles of CAR can be extended to other AR frameworks. By focusing on the spatial positioning of tokens, we envision that similar conditioning strategies can be implemented. It is important to recognize that CAR represents a significant step forward in the field of AR model control, particularly in maintaining the original model's capabilities. We will explore these possibilities in future work.
>
> **4**. As the performance of smaller CAR modules with larger pretrained VAR models, it is a interesting question. That may work, we consider it needs either initialize control modules from smaller VAR for larger VAR, or just using less copy parameters from original models. We would like to have such a try in the future, again thanks for your advice.
>
> We sincerely thank you once again for your thorough review and constructive feedback. We are eager to incorporate your suggestions to improve the quality and impact of our work.

---

### Author Response · Authors · 2024-11-15
**Withdrawal & Response to Reviewer Comments**

Thank you all for your thorough review, which will surely help us doing better. From the readers' perspective, it is quite reasonable to have these questions. We understand that there are areas in our manuscript where the explanations could be clearer, and we recognize that we have not sufficiently highlighted the unique contributions of this work.

We would like to consider this work an insightful validation of the feasibility and specific methods for controlling AR models using an approach akin to ControlNet in diffusion models. Given the significant success of ControlNet in the diffusion domain, we believe that the broader community would find inspiration in a similar control solution for AR models, which is at the core of our contribution.

However, despite our belief that we have the opportunity to address all the questions raised, the initial rating scores we received were rather low. We feel that the chances of a successful rebuttal are slim. Nonetheless, we would like to provide basic responses to your questions below as a friendly communication.

---

### Note · Authors · 2024-11-15

I have read and agree with the venue's withdrawal policy on behalf of myself and my co-authors.